# Obesity and mortality after the first ischemic stroke: Is obesity paradox real?

**Durgesh Chaudhary**[1], **Ayesha Khan**[1], **Mudit Gupta**[2], **Yirui Hu**[3], **Jiang Li**[4], **Vida Abedi**[4,5], **Ramin Zand**[1] *

**1** Geisinger Neuroscience Institute, Geisinger Health System, Danville, Pennsylvania, United States of America, **2** Phenomic Analytics and Clinical Data Core, Geisinger Health System, Danville, Pennsylvania, United States of America, **3** Department of Population Health Sciences, Geisinger Health System, Danville, Pennsylvania, United States of America, **4** Department of Molecular and Functional Genomics, Weis Center for Research, Geisinger Health System, Danville, Pennsylvania, United States of America, **5** Biocomplexity Institute, Virginia Tech, Blacksburg, Virginia, United States of America

* ramin.zand@gmail.com, rzand@geisinger.edu

## Abstract

**Data Availability Statement:** Comprehensive summary and aggregate data have been included in the manuscript and its  supporting information files. The patient-level data analyzed in this study is not publicly available due to privacy concerns.

### Background and purpose

Obesity is an established risk factor for ischemic stroke but the association of increased body mass index (BMI) with survival after ischemic stroke remains controversial. Many studies have shown that increased BMI has a "protective" effect on survival after stroke while other studies have debunked the "obesity paradox". This study aimed at examining the relationship between BMI and all-cause mortality at one year in first-time ischemic stroke patients using a large dataset extracted from different resources including electronic health records.

### Methods

This was a retrospective cohort study of consecutive ischemic stroke patients captured in our Geisinger NeuroScience Ischemic Stroke (GNSIS) database. Survival in first-time ischemic stroke patients in different BMI categories was analyzed using Kaplan Meier survival curves. The predictors of mortality at one-year were assessed using a stratified Cox proportional hazards model.

### Results

Among 6,703 first-time ischemic stroke patients, overweight and obese patients were found to have statistically decreased hazard ratio (HR) compared to the non-overweight patients (overweight patients- HR = 0.61 [95% CI, 0.52–0.72]; obese patients- HR = 0.56 [95% CI, 0.48–0.67]). Predictors with a significant increase in the hazard ratio for one-year mortality were age at the ischemic stroke event, history of neoplasm, atrial fibrillation/flutter, diabetes, myocardial infarction and heart failure.

Patient-level data access requests should be addressed to the Geisinger Institutional Review Board (irb@geisinger.edu) or the corresponding author.

**Funding:** This study had no specific funding. VA had financial research support from the Defense Threat Reduction Agency (DTRA) grant No. HDTRA1-18-1-0008 sub-awarded to Geisinger and funds from the National Institute of Health (NIH) grant No. R56HL116832 sub-awarded to Geisinger during the study period. RZ had financial research support from Bucknell University Initiative Program, Roche – Genentech Biotechnology Company, the Geisinger Health Plan Quality fund during the study period, and receives institutional support from Geisinger Health System. The funders had no role in study design, data collection and analysis, or preparation of the manuscript.

**Competing interests:** The authors have read the journal's policy and declare the following competing interests: RZ had financial research support from Bucknell University Initiative Program, Roche – Genentech Biotechnology Company. There are no patents, products in development or marketed products associated with this research to declare. This does not alter our adherence to PLOS ONE policies on sharing data and materials

## Conclusion

Our study results support the obesity paradox in ischemic stroke patients as shown by a significantly decreased hazard ratio for one-year mortality among overweight and obese patients in comparison to non-overweight patients.

## Introduction

Stroke ranks fifth among the leading causes of mortality with 1 in 19 deaths in the United States [1]. Obesity has been recognized as an important risk factor for cardiovascular and cerebrovascular diseases. Increased body mass index (BMI) is associated with shorter life expectancy and morbidity-free life [2].

While obesity is an established independent risk factor for stroke [3, 4], the association of increased BMI and survival after stroke remains controversial. There is a growing consensus for the inverse relationship between obesity and outcome after cardiovascular diseases and stroke [5]. Many studies on stroke mortality and stroke recurrence have reported favorable outcomes for stroke patients with higher BMI [6–12] while other studies have concluded that the so-called "obesity paradox" does not exist [13] or demonstrated no survival advantage for obesity in cardiovascular diseases [14]. The 'Guidelines for the Prevention of Stroke in Patients with Stroke and Transient Ischemic Attack' (2014) by American Heart Association/American Stroke Association, recommends screening all transient ischemic attack or stroke patients for obesity with BMI measurement but mentions that the benefits of weight reduction among obese patients after recent TIA or stroke is uncertain [15]. The proposed mechanisms behind the obesity paradox are not established and there could be unknown factors and confounders contributing to it.

The goal of this study was to examine the association of BMI and all-cause mortality among a cohort of first-time ischemic stroke patients in Pennsylvania, United States.

## Methods

### Database description and processing

This is a retrospective cohort study based on extracted data from multiple resources including Geisinger's Electronic Health Record (EHR) system, Geisinger Quality database, Geisinger Health Plan claims data, as well as the Social Security Death database to build a stroke database called "Geisinger NeuroScience Ischemic Stroke (GNSIS)". Geisinger is a fully integrated health system that serves central, south-central, and northeast Pennsylvania. GNSIS includes demographic, clinical, laboratory and imaging data from 8,929 ischemic stroke patients from September 2003 to May 2019. The study was reviewed and approved by the Geisinger Institutional Review Board (IRB No. 2019–0470). Informed consent was waived by the Geisinger Institutional Review Board as the study only utilized deidentified data.

The GNSIS database was created based on a high-fidelity and data-driven phenotype definition for ischemic stroke developed by our team at Geisinger. The patients were included in the GNSIS database if they had a primary discharge diagnosis of ischemic stroke; a brain magnetic resonance imaging (MRI) during the same encounter to confirm the diagnosis; and an overnight stay in the hospital. The diagnosis of ischemic stroke and other comorbidities were based on International Classification of Diseases, Ninth/Tenth Revision, Clinical Modification (ICD-9-CM/ICD-10-CM) codes (Table A in S1 Text). The Current Procedural Terminology

(CPT)-4 codes for brain MRIs are available in Table B in S1 Text. The following data elements were recorded in the GNSIS database: date of the event, age of the patient at index stroke, encounter type, ICD-9-CM/ICD-10-CM code and corresponding primary diagnosis of index stroke, the National Institutes of Health Stroke Scale (NIHSS) score, presence or absence (and if applicable date) of recurrent stroke, and ICD-9-CM/ICD-10-CM code and corresponding primary diagnosis of the recurrent stroke. Other data elements include sex, birth date, death date, presence or absence of atrial fibrillation, atrial flutter, hypertension, myocardial infarction, diabetes mellitus, hypercoagulable states, chronic liver diseases, dyslipidemia, heart failure, chronic lung diseases (chronic obstructive pulmonary disease, asthma, occupational lung diseases), chronic kidney diseases, peripheral vascular diseases, rheumatic diseases, and neoplasms, family history of heart disorders or stroke, prior history of ischemic stroke or hemorrhagic stroke, and smoking status (Table C in S1 Text). In the case of multiple encounters due to recurrent cerebral infarcts, the first hospital encounter was considered as the index stroke. To ensure that patients were active, the last encounter of patients was also recorded. The NIHSS was extracted from the quality data and matched with EHR data using medical record numbers. To ensure the comprehensiveness of follow up information, data from all encounter types as well as insurance claims data were pulled and processed. Furthermore, our database interfaces with the Social Security Death Index on a biweekly basis to reflect updated information on the vital status. GNSIS database has been described in a previous paper [16].

As part of data pre-processing, several steps were taken to ensure data integrity and validity. Units were verified and reconciled if needed and distributions of variables were assessed over time to ensure data stability. The range for the variables was defined according to expert knowledge and available literature—and outliers were assessed and replaced or capped based on the other clinical data. As a part of the de-identification process, the age of patients older than 89 years old was masked and changed to 89 and thus the date of death was approximated. Filters were applied to ensure that the relevant variables were captured within the desired timeframe.

## Cohort and outcome definitions

For this study, only first-time ischemic stroke patients were selected for analysis. We excluded patients based on the following: 1) patients who had a previous history of stroke outside of Geisinger 2) patients younger than 18 years at the time of ischemic stroke, and 3) patients with no baseline BMI recorded in the EHR. All-cause mortality at one year was defined as death due to any cause within one year of index stroke date as recorded in the database.

Thirty-three demographic/clinical variables were used in this study. The variables NIHSS and BMI had some missingness. NIHSS was available for 1,782 patients and BMI was missing for 852 adult patients of the GNSIS cohort after excluding patients with a previous history of stroke. Both the NIHSS and BMI were examined to determine if the missingness were at random.

We grouped patients in four different categories based on their baseline BMI. The latter was calculated from the height and weight extracted from the EHR within the past three years of the ischemic stroke event and median BMI during this period was taken as the baseline BMI. Per the World Health Organization (WHO) definitions, patients were categorized as underweight if their BMI was less than 18.5 kg/m$^2$. Patients with a BMI of 18.5 to 24.9 kg/m$^2$ and BMI of 25 to 29.9 kg/m$^2$ were categorized as normal weight and overweight, respectively. Patients with a BMI of 30 kg/m$^2$ and above were categorized as obese. The non-overweight category was defined by merging underweight and normal weight categories.

## Statistical analysis

The patients were compared regarding their baseline characteristics, medical history before ischemic stroke event, and stroke severity using NIHSS. The categorical and continuous variables were compared using the chi-square test and analysis of variance (ANOVA) test, respectively. Continuous variables were presented as mean ± standard deviation (normal distribution) and as median with interquartile range (non-normal distribution).

Kaplan-Meier estimator was used to estimating the probability of survival at 30, 90, and 365 days from the date of the ischemic stroke event. The log-rank test was used to determine the statistical significance of the differences in the survival curves of the different BMI categories. To examine the predictors of all-cause mortality after an ischemic stroke event, the stratified Cox proportional hazard model was used. All factors with a p-value of < 0.1 in univariate analysis were included in the stratified Cox model. Potential predictors (p < 0.1 in univariate analysis) not satisfying the proportional hazards assumptions were used as strata in the model. The proportional hazards assumption was assessed by the Schoenfeld test. A multivariate logistic regression was also used in the subset of stroke patients who had at least one-year follow-up data and odds-ratio (OR) were given. All statistical analyses were performed in R (version 3.6.2) [17], RStudio (version 1.2.5019) [18] using "survival" [19] and "survminer" [20] packages.

## Results

A total of 8,929 patients were identified and extracted from the GNSIS database, where 24.9% (2,226) of the patients met the exclusion criteria (Fig A in S1 Text). There was no significant difference between patients with missing BMI (852 adult patients) and the rest of the cohort in terms of median age, gender distribution, and NIHSS (Table D and Fig B in S1 Text).

Out of 6,703 patients included in the study, 51.9% were men and 95.0% were Caucasians. The median age was 71.5 years (interquartile range, IQR = 61.0–81.2). In the one-year follow-up period, 917 (13.7%) deaths were recorded. The mean BMI of the patients was 29.5 (SD 6.6) kg/m$^2$. A total of 115 (1.7%) patients were underweight, 1,523 (22.7%) were in normal weight category, 2,298 (34.3%) were overweight and the remaining 2,767 (41.3%) were obese. The patients' demographics, past medical history, and family history are summarized in Table 1.

### Higher survival correlates with higher BMI

Survival curve in the different BMI groups until one year after ischemic stroke is shown in Fig 1A. While the global p-value for the log-rank test was less than 0.0001, pairwise log-rank test with p-value adjustment showed that survival was significantly different between all BMI groups (all p < 0.01) except between underweight and normal BMI categories (p = 0.9892, Table E in S1 Text).

When female and male patients were considered separately (Fig 1B and 1C), the differences in the survival curves were still significant (p < 0.0001). The pairwise log-rank test revealed that survival in the underweight patients was not statistically different from normal BMI in both male and female patients. While pairwise log-rank tests for male patients mirrored results shown for the overall cohort (only non-significant p-value between underweight and normal weight category), the same was not true for female patients. Among women, only significant differences were observed between normal and overweight patients, and normal and obese patients (Table E in S1 Text).

As the underweight and normal BMI categories showed a non-significant difference in survival at one year and the number of patients was small in the underweight category (115 patients), these two categories were merged to form the non-overweight category. Survival at

**Table 1. Patient demographics, past medical and family history.**

| Patient Characteristics | Overall | Underweight (BMI <18.5 kg/m$^2$) | Normal (BMI 18.5 to 24.9 kg/m$^2$) | Overweight (BMI 25.0 to 29.9 kg/m$^2$) | Obese (BMI ≥30.0 kg/m$^2$) | p-value |
|---|---|---|---|---|---|---|
| Total number of patients | 6,703 | 115 | 1,523 | 2,298 | 2,767 | |
| Age in years, mean (SD) | 70.2 (13.5) | 75.2 (14.4) | 73.7 (13.9) | 71.5 (13.2) | 67.1 (12.7) | <0.001 |
| Age in years, median [IQR] | 71.5 [61.0, 81.2] | 79.1 [64.5, 86.3] | 77.1 [64.5, 85.6] | 73.4 [63.0, 82.0] | 67.9 [58.8, 76.8] | <0.001 |
| Male, n (%) | 3,476 (51.9) | 28 (24.3) | 691 (45.4) | 1,341 (58.4) | 1,416 (51.2) | <0.001 |
| Body mass index (BMI) in kg/m$^2$, mean (SD) | 29.5 (6.6) | | | | | |
| Body mass index (BMI) in kg/m$^2$, median [IQR] | 28.6 [25.0, 32.9] | | | | | |
| Current smoker, n (%) | 1,048 (15.6) | 26 (22.6) | 287 (18.8) | 351 (15.3) | 384 (13.9) | <0.001 |
| National Institute of Health Stroke Scale (NIHSS)*, median [IQR] | 3 [2, 6] | 4 [2, 10] | 4 [2, 8] | 3 [2, 6] | 3 [2, 6] | <0.001 |
| Ischemic stroke recurrence within 1 year, n (%) | 303 (4.5) | 2 (1.7) | 70 (4.6) | 115 (5.0) | 116 (4.2) | 0.259 |
| All-cause mortality at 1 year, n (%) | 917 (13.7) | 24 (20.9) | 324 (21.3) | 289 (12.6) | 280 (10.1) | <0.001 |
| Past Medical History | | | | | | |
| Atrial fibrillation or flutter, n (%) | 1,440 (21.5) | 27 (23.5) | 348 (22.8) | 494 (21.5) | 571 (20.6) | 0.371 |
| Hypertension, n (%) | 5,260 (78.5) | 70 (60.9) | 1,115 (73.2) | 1,772 (77.1) | 2,303 (83.2) | <0.001 |
| Myocardial infarction, n (%) | 763 (11.4) | 13 (11.3) | 162 (10.6) | 262 (11.4) | 326 (11.8) | 0.734 |
| Diabetes, n (%) | 2,236 (33.4) | 12 (10.4) | 297 (19.5) | 687 (29.9) | 1240 (44.8) | <0.001 |
| Dyslipidemia, n (%) | 4,380 (65.3) | 43 (37.4) | 878 (57.6) | 1,529 (66.5) | 1,930 (69.8) | <0.001 |
| Heart failure, n (%) | 900 (13.4) | 18 (15.7) | 202 (13.3) | 277 (12.1) | 403 (14.6) | 0.062 |
| Hypercoagulable states, n (%) | 88 (1.3) | 1 (0.9) | 25 (1.6) | 24 (1.0) | 38 (1.4) | 0.423 |
| Chronic liver disease, n (%) | 205 (3.1) | 5 (4.3) | 41 (2.7) | 58 (2.5) | 101 (3.7) | 0.078 |
| Chronic lung diseases, n (%) | 1,538 (22.9) | 44 (38.3) | 383 (25.1) | 474 (20.6) | 637 (23.0) | <0.001 |
| Rheumatic diseases, n (%) | 287 (4.3) | 1 (0.9) | 87 (5.7) | 90 (3.9) | 109 (3.9) | 0.006 |
| Chronic kidney disease, n (%) | 1,293 (19.3) | 18 (15.7) | 270 (17.7) | 417 (18.1) | 588 (21.3) | 0.007 |
| Neoplasm, n (%) | 1,119 (16.7) | 20 (17.4) | 286 (18.8) | 433 (18.8) | 380 (13.7) | <0.001 |
| Peripheral vascular disease, n (%) | 1,094 (16.3) | 23 (20.0) | 268 (17.6) | 414 (18.0) | 389 (14.1) | <0.001 |
| Family History | | | | | | |
| Family history of heart disease, n (%) | 2,505 (37.4) | 36 (31.3) | 478 (31.4) | 838 (36.5) | 1,153 (41.7) | <0.001 |
| Family history of stroke, n (%) | 956 (14.3) | 11 (9.6) | 223 (14.6) | 318 (13.8) | 404 (14.6) | 0.415 |

*NIHSS data available for 1,782 patients (28 underweight, 427 normal weight, 628 overweight and 699 obese patient).

one year was re-analyzed utilizing three BMI categories for the overall cohort (Fig 2A) and female and male patients separately (Fig 2B and 2C). The survival at one year was significantly different between all BMI categories for the entire cohort and male patients (all p-values in pairwise log-rank tests < 0.05) with better survival for higher BMI categories. For female patients, survival was significantly higher in overweight and obese categories compared to non-overweight categories but there was no significant difference between overweight and obese patients (pairwise log-rank test p = 0.086; Table F in S1 Text).

The cumulative survival probability at 30, 90, and 365 days after first-time ischemic stroke for different BMI categories is shown in Table 2. Survival probabilities at 30, 90 and 365 days were significantly higher in overweight and obese categories compared to the non-overweight category but there was no significant difference between the overweight and obese category.

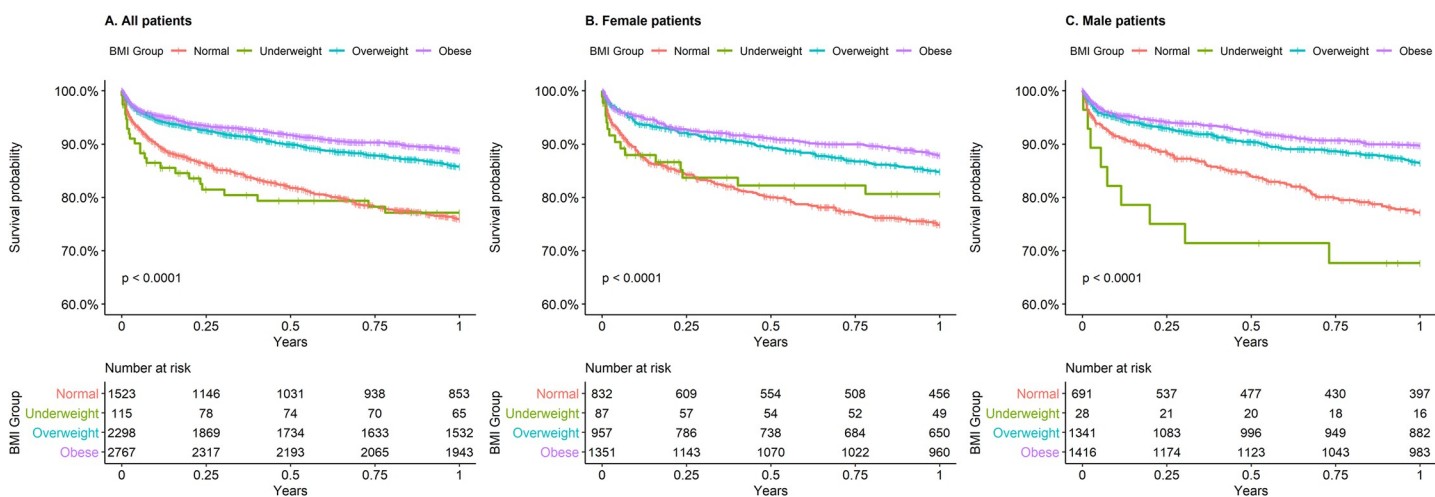

**Fig 1. Kaplan-Meier survival curves for all-cause mortality stratified by four BMI groups.** A, All patients. B, Female patients. C, Male patients.

### Reduced hazard ratio for one-year mortality in overweight and obese patients

In the stratified Cox proportional hazards model, overweight and obese patients were found to have statistically decreased hazard ratio (HR) compared to the non-overweight patients (overweight patients- HR = 0.61 [95% CI, 0.52–0.72]; obese patients- HR = 0.56 [95% CI, 0.48–0.67]). Male patients had significantly lower HR than female patients in univariate analysis, but it became non-significant in the stratified Cox model. Predictors with a significant increase in HR for one-year mortality were age at an ischemic stroke, as well as the history of neoplasm, atrial fibrillation/flutter, diabetes, myocardial infarction and heart failure. The results of the univariate analysis and the stratified Cox model are shown in Table 3.

### Reduced hazard ratio in overweight and obese patients persists when considering stroke severity

The NIHSS was available for 1,782 patients included in this study. Another stratified Cox model was employed for this subset of patients with NIHSS as one of the strata. The hazard

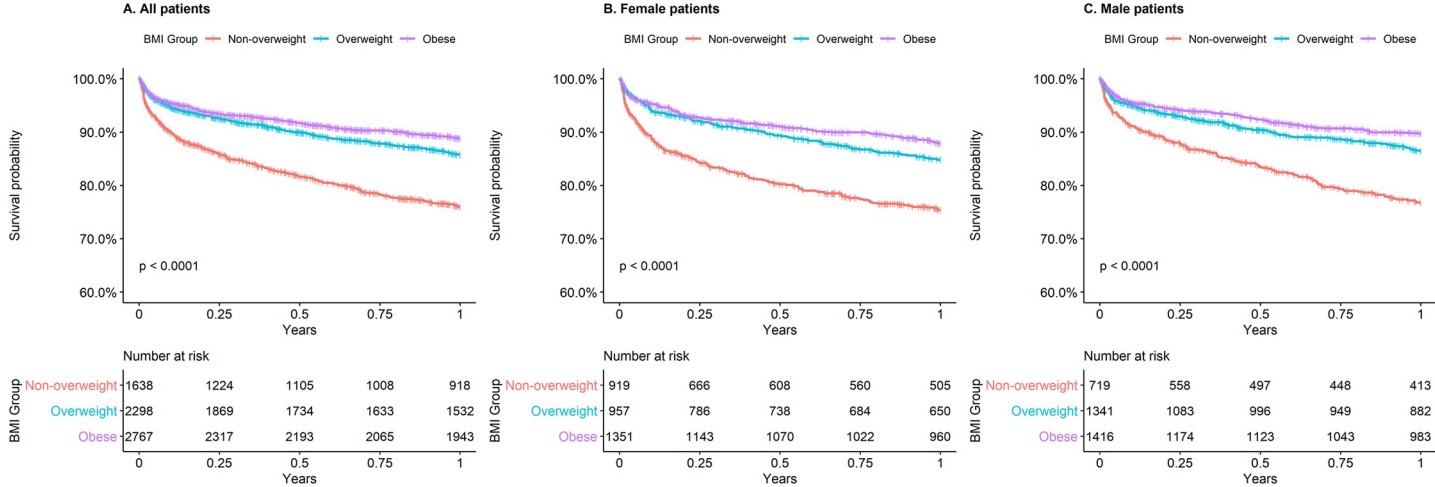

**Fig 2. Kaplan-Meier survival curves for all-cause mortality stratified by three BMI groups.** A, All patients. B, Female patients. C, Male patients.

**Table 2. Cumulative probability of survival in first-time ischemic stroke patients by BMI category.**

| | Cumulative probability of survival, % (95% CI) | | | p-value |
|---|---|---|---|---|
| | **Non-overweight** | **Overweight** | **Obese** | |
| **At 30 days** | 91.0 (89.6–92.4) | 95.4 (94.5–96.2) | 95.9 (95.2–96.7) | <0.0001 |
| **At 90 days** | 86.8 (85.2–88.4) | 93.0 (92.0–94.1) | 93.8 (92.9–94.7) | <0.0001 |
| **At 1 year** | 78.1 (76.0–80.1) | 87.0 (85.6–88.4) | 89.5 (88.4–90.7) | <0.0001 |

ratio for one-year mortality in overweight and obese patients were still found to be significantly lower than non-overweight patients (overweight patients- HR = 0.58 [95% CI, 0.41–0.81]; obese patients- HR = 0.51 [95% CI, 0.35–0.73]). In this subset of patients, age at ischemic stroke event, as well as the history of heart failure, neoplasm, and hypercoagulable states were associated with a significantly increased hazard ratio. Table G in S1 Text includes the details of the stratified Cox model for patients with NIHSS data. (For assessment of proportional hazards assumption, see Tables H–I in S1 Text.)

## Multivariate logistic regression in subset of patients with at least one-year of follow-up

A total 6,015 ischemic stroke patients either had one-year of follow-up data or died within one year. When multivariate logistic regression was performed in this subset of patients, overweight and obese patients showed decreased risk of one-year mortality compared to non-overweight patients (overweight patients- OR = 0.54 [95% CI, 0.45–0.65]; obese patients-

**Table 3. Univariate Cox analysis and multivariate stratified Cox proportional hazards model on one-year mortality.**

| | Univariate Analysis | | | Multivariate Stratified Cox Model | | |
|---|---|---|---|---|---|---|
| **Variable** | **Hazard Ratio** | **95% CI** | **p-value** | **Hazard Ratio** | **95% CI** | **p-value** |
| BMI categories | | | | | | |
| Non-overweight | Reference | | | Reference | | |
| Overweight | 0.55 | (0.48–0.65) | < 0.01 | 0.61 | (0.52–0.72) | < 0.01 |
| Obese | 0.44 | (0.38–0.52) | < 0.01 | 0.56 | (0.48–0.67) | < 0.01 |
| Male (vs. Female) | 0.83 | (0.73–0.94) | < 0.01 | 1.02 | (0.89–1.17) | 0.786 |
| Age at ischemic stroke | 1.05 | (1.05–1.06) | < 0.01 | 1.04 | (1.03–1.04) | < 0.01 |
| Myocardial infarction | 1.69 | (1.42–2.00) | < 0.01 | 1.23 | (1.02–1.48) | 0.032 |
| Neoplasm | 2.19 | (1.9–2.53) | < 0.01 | 1.59 | (1.37–1.84) | < 0.01 |
| Atrial fibrillation or flutter | 2.04 | (1.78–2.34) | < 0.01 | 1.26 | (1.09–1.46) | < 0.01 |
| Heart Failure | 2.74 | (2.37–3.16) | < 0.01 | 1.67 | (1.42–1.98) | < 0.01 |
| Chronic lung diseases | 1.24 | (1.07–1.43) | < 0.01 | 1.05 | (0.9–1.22) | 0.539 |
| Diabetes | 1.2 | (1.06–1.38) | < 0.01 | 1.26 | (1.09–1.46) | < 0.01 |
| Peripheral vascular disease | 1.48 | (1.27–1.73) | < 0.01 | * | | |
| Chronic kidney disease | 2.24 | (1.95–2.57) | < 0.01 | * | | |
| Dyslipidemia | 0.82 | (0.72–0.94) | < 0.01 | * | | |
| Rheumatic diseases | 1.64 | (1.27–2.13) | < 0.01 | * | | |
| Hypertension | 1.12 | (0.95–1.32) | 0.16 | † | | |
| Chronic liver disease | 1.16 | (0.81–1.66) | 0.42 | † | | |
| Hypercoagulable states | 1.15 | (0.68–1.95) | 0.61 | † | | |

*Variables included as strata in the stratified Cox model
†Variables not included in the stratified Cox model.

OR = 0.50 [95% CI, 0.41–0.60]). The details of the multivariate logistic regression are included in Table J in S1 Text.

## Discussion

Although studies have indicated that overweight patients are at increased risk of ischemic stroke [4], there has been evidence suggesting that the risk of mortality might be lower among these patients. A meta-analysis on BMI and its association with mortality and recurrent stroke in stroke patients supported this "protective" effect [21]. Another systematic review on the obesity paradox in stroke has indicated increased survival of obese stroke patients after stroke incident but also raised concerns about the methodologies of the reviewed studies [22]. Obesity paradox has also been observed in other diseases including peripheral vascular disease, coronary artery disease, and atrial fibrillation [23–25].

In contrast, a study on BMI and stroke deaths did not find any obesity paradox [13]. A collaborative analysis of baseline BMI and mortality in 57 prospective studies showed a 40% higher vascular mortality for each 5 kg/m$^2$ increase in BMI [26]. Another study found a U-shaped relationship between BMI and mortality [27]. A more recent study found that being overweight or obese did not translate to decreased mortality but being underweight was associated with unfavorable outcomes [28].

Our study supports the studies [29–33] which have shown that patients with higher BMI have lower short-term and long-term mortality after stroke. In our study, the hazard ratio associated with all-cause mortality decreased significantly in the overweight and obese category compared to the non-overweight category after adjusting for age at ischemic stroke event, sex, baseline comorbidities, and stroke severity. Our result in the subset of patients with NIHSS data is different from that by Ryu *et al.* [34] which showed that after adjustment of initial neurological severity using NIHSS, the hazard ratio for underweight patients remained significant while the favorable outcome in overweight and obese patients did not. Different studies on the obesity paradox are summarized in Table K of S1 Text.

Some possible mechanisms behind the obesity paradox have been explored. One of the possible explanations for the obesity paradox could be hypoxic preconditioning (HPC). HPC refers to resistance to severe hypoxia and tissue injury after exposure to moderate and intermittent hypoxia and is well documented in animal models [35, 36]. The effect of preconditioning due to chronic intermittent hypoxia in obese patients may play a role in explaining obesity paradox [37, 38], but the underlying mechanisms still remain largely unexplored. We explored obstructive sleep apnea (OSA) rates in our study participants. OSA rate in non-overweight, overweight, and obese patients was 1.4%, 3.1% and 11.5% respectively. OSA was not associated with one-year all-cause mortality and its inclusion in the stratified cox model did not change our results.

Stroke can cause increased systemic neuroendocrine imbalances, pro-inflammatory cytokines, radical overload leading to a catabolic state, and thus patients with a higher metabolic reserve are more likely to have a better outcome [39, 40]. A potentially protective effect of adipose tissue could be due to its secretion of soluble TNF-α receptors and thus neutralizing the effect of TNF-α [41]. Another possible reason could be the cardiorespiratory fitness status of patients regardless of the BMI. In a study by Ortega et al., "metabolically healthy obese" patients had significantly better fitness and lower risk for all-cause mortality compared to unhealthy obese patients [42].

Obesity paradox has been observed mostly when using BMI as an index for obesity. BMI has limitations in differentiating lean body mass from fat body mass. Further, BMI also does not consider the distribution/location of fat in the body. Indeed, the obesity paradox has not been

observed when using indices like waist-to-hip ratio and body fat percentage [43, 44]. It could also be that most overweight and obese patients die relatively younger and the patients included in these studies are simply a healthier subset of the overweight or obese population [27].

The presented study has several strengths and limitations. The use of EHR and other data resources (quality, claims, and the social security data) provided a comprehensive overview of patients' clinical data with deep phenotype and clinical evaluation as well as a large sample size. Additionally, results from studies using EHR can be easily translated into clinical practice when fully validated. Furthermore, Geisinger has a stable patient population and a rich longitudinal dataset. However, using these data resources has its drawbacks, mainly centered around inherent noise due to the nature of the data, as well as biased patient selection. Patients were included in the GNSIS database based on three criteria of ICD-9-CM/ICD-10-CM code, brain MRI and encounter duration and thus, cases of ischemic stroke not fulfilling these criteria might be missing from this database. Baseline BMI was also missing for approximately 10% of patients in the GNSIS database. These were determined to be missing at random in our analysis and excluded from the study. Similarly, the baseline BMI of the patients is not from the same time with respect to the stroke event as it was calculated from a three-year window. Waist-to-hip ratio data was highly sparse in the structured EHR data limiting its utility in this study. Furthermore, the NIHSS score was only available for 1,782 patients (28 underweight, 427 normal weight, 628 overweight, and 699 obese patients). Our dataset was also imbalanced with respect to the underweight group. However, our main findings remained unchanged after merging underweight and normal-weight groups. Finally, we did not perform a deep phenotyping evaluation (by chart review) due to the size of the dataset, which might have helped us identify other important clinical features that were not captured by our data extraction process. The predominantly Caucasian population of the GNSIS database may be a concerning factor limiting the generalization of results to a diverse population.

We considered the patient's age, gender, comorbidities, and initial NIHSS at the time of first-time stroke inclusion in the stratified Cox models in our study. New diagnoses of comorbidities and lifestyle modifications like exercising, changes in alcohol consumption or smoking, dietary changes, social support after stroke events may play an important role in a patient's survival and outcome. Our analysis revealed that the rate of new diagnoses of comorbidities in our study population after stroke event was low. Except for chronic lung diseases, there was no significant difference between patients in different BMI categories in terms of newly developed comorbidities (Tables L–M in S1 Text). Changes in lifestyle modification are poorly documented in the EHR and were excluded in the study.

## Conclusion

Our study results support the obesity paradox in ischemic stroke patients as shown by a significantly decreased hazard ratio for one-year mortality among overweight and obese patients in comparison to non-overweight patients. Despite the various studies confirming the obesity paradox, there is no change in the recommendations regarding weight loss in cardiovascular and cerebrovascular patients. The obesity paradox would have to be confirmed beyond any doubt before changes in these recommendations can be proposed. The actual biological process through which obesity provides this "protective" effect on mortality, as well as potential confounders for this phenomenon, should be investigated.

## Supporting information

**S1 Text. Supplemental materials.**
(DOCX)

## Author Contributions

**Conceptualization:** Vida Abedi, Ramin Zand.

**Data curation:** Mudit Gupta.

**Formal analysis:** Durgesh Chaudhary, Yirui Hu, Jiang Li, Vida Abedi, Ramin Zand.

**Investigation:** Durgesh Chaudhary.

**Methodology:** Yirui Hu, Jiang Li, Vida Abedi, Ramin Zand.

**Project administration:** Vida Abedi.

**Resources:** Ramin Zand.

**Supervision:** Vida Abedi, Ramin Zand.

**Validation:** Ayesha Khan.

**Visualization:** Durgesh Chaudhary.

**Writing – original draft:** Durgesh Chaudhary.

**Writing – review & editing:** Ayesha Khan, Mudit Gupta, Yirui Hu, Jiang Li, Vida Abedi, Ramin Zand.

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
