## [Decision Letter · Decision Letter 0]

5 Jan 2021

PONE-D-20-39440

Obesity and Mortality after the First Ischemic Stroke: Is Obesity Paradox Real?

PLOS ONE

Dear Dr. Zand,

Thank you for submitting your manuscript to PLOS ONE. After careful consideration, we feel that it has merit but does not fully meet PLOS ONE’s publication criteria as it currently stands. Therefore, we invite you to submit a revised version of the manuscript that addresses the points raised during the review process.

We look forward to receiving your revised manuscript.

Kind regards,

Aristeidis H. Katsanos, MD, PhD

Academic Editor

PLOS ONE

Journal Requirements:

2) We note that you have indicated that data from this study are available upon request. PLOS only allows data to be available upon request if there are legal or ethical restrictions on sharing data publicly. For information on unacceptable data access restrictions, please see http://journals.plos.org/plosone/s/data-availability#loc-unacceptable-data-access-restrictions.

3) Thank you for stating the following in the Acknowledgments Section of your manuscript:

[This study had no specific funding. VA had financial research support from the Defense Threat Reduction Agency (DTRA) grant No. HDTRA1-18-1-0008 sub-awarded to Geisinger and funds from the National Institute of Health (NIH) grant No. R56HL116832 sub-awarded to Geisinger during the study period. RZ had financial research support from Bucknell University Initiative Program, Roche – Genentech Biotechnology Company, the Geisinger Health Plan Quality fund, and receives institutional support from Geisinger Health System during the study period.]

 [The author(s) received no specific funding for this work.]

Additionally, because some of your funding information pertains to [commercial funding//patents], we ask you to provide an updated Competing Interests statement, declaring all sources of commercial funding.

In your Competing Interests statement, please confirm that your commercial funding does not alter your adherence to PLOS ONE Editorial policies and criteria by including the following statement: "This does not alter our adherence to PLOS ONE policies on sharing data and materials.” as detailed online in our guide for authors  http://journals.plos.org/plosone/s/competing-interests.  If this statement is not true and your adherence to PLOS policies on sharing data and materials is altered, please explain how.

Please include the updated Competing Interests Statement and Funding Statement in your cover letter. We will change the online submission form on your behalf.

4) PLOS requires an ORCID iD for the corresponding author in Editorial Manager on papers submitted after December 6th, 2016. Please ensure that you have an ORCID iD and that it is validated in Editorial Manager. To do this, go to ‘Update my Information’ (in the upper left-hand corner of the main menu), and click on the Fetch/Validate link next to the ORCID field. This will take you to the ORCID site and allow you to create a new iD or authenticate a pre-existing iD in Editorial Manager. Please see the following video for instructions on linking an ORCID iD to your Editorial Manager account: https://www.youtube.com/watch?v=_xcclfuvtxQ

5) Please include captions for your Supporting Information files at the end of your manuscript, and update any in-text citations to match accordingly. Please see our Supporting Information guidelines for more information: http://journals.plos.org/plosone/s/supporting-information.

Reviewers' comments:

Reviewer's Responses to Questions

**Comments to the Author**

1. Is the manuscript technically sound, and do the data support the conclusions?

Reviewer #1: Yes

Reviewer #2: Yes

2. Has the statistical analysis been performed appropriately and rigorously? 

Reviewer #1: Yes

Reviewer #2: Yes

3. Have the authors made all data underlying the findings in their manuscript fully available?

Reviewer #1: Yes

Reviewer #2: No

4. Is the manuscript presented in an intelligible fashion and written in standard English?

Reviewer #1: Yes

Reviewer #2: Yes

5. Review Comments to the Author

Reviewer #1: This is a retrospective, cohort study of consecutive first-ever ischemic stroke patients with the aim to investigate the association between baseline BMI and all-cause mortality after stroke. This is a well-designed study, based on high quality data derived from GNSIS database, assessed with robust statistical analyses, producing well-presented results supporting the obesity paradox which has been questioned in previous studies. What is more, the authors have already adequately discussed all potential limitations.

For further discussion and due to the existence of data of such a high quality, it would be of interest if an association could be presented between the obesity and mortality among specific ischemic stroke subtypes.

Reviewer #2: The study by Chaudhary et al. provides evidence in support of the “obesity paradox” in ischemic stroke patients. In this retrospective cohort of 6,703 first-time ischemic stroke patients, overweight and obese patients were found to have statistically decreased hazard ratio for one-year mortality in comparison to non-overweight patients. The manuscript is very well written, the methods and statistical analyses are adequately presented. The authors could consider few suggestions that might further improve the results/discussion.

1. Are underlying stroke etiologies captured in the GNSIS database? Considering the noted differences in cardiovascular risk factors in obese/overweight vs. non-overweight patients (as shown in Table 1), it would be interesting to assess whether small vessel disease or other underlying stroke etiologies might differ between obese/overweight vs. non-overweight patients (i.e., with subsequent implications in the functional recovery after stroke).

2. Could the authors discuss on whether ischemic/hypoxic preconditioning might be associated with improved functional outcome in obese/overweight patients? Can the rates of concomitant Obesity hypoventilation syndrome (OHS) be retrieved for included patients?

Minor

- Line 256, p 12: “has” should be corrected to “have”.

6. PLOS authors have the option to publish the peer review history of their article (what does this mean?). If published, this will include your full peer review and any attached files.

Reviewer #1: No

Reviewer #2: **Yes: **Maria-Ioanna Stefanou

---

## [Author Response · Author response to Decision Letter 0]

26 Jan 2021

We would like to thank you for inviting us to submit a revised draft of our manuscript titled “Obesity and mortality after the first ischemic stroke: Is obesity paradox real?” to PLOS ONE.

We would also like to thank you and the reviewers for your time and effort to provide us with your valuable feedback on this paper. We have made changes to the manuscript to reflect the suggestions that you have provided. We hope our edits and responses below are able to address your concerns.

Journal Requirements:

Response: The manuscript has been edited to meet PLOS ONE’s style requirements. A revised cover letter has been submitted to address data availability, funding statement and competing interests statement. ORCID iD is available for the corresponding author. Captions for Supporting Information files have been added at the end of the manuscript.

Reviewer 1 comments:

This is a retrospective, cohort study of consecutive first-ever ischemic stroke patients with the aim to investigate the association between baseline BMI and all-cause mortality after stroke. This is a well-designed study, based on high quality data derived from GNSIS database, assessed with robust statistical analyses, producing well-presented results supporting the obesity paradox which has been questioned in previous studies. What is more, the authors have already adequately discussed all potential limitations.

For further discussion and due to the existence of data of such a high quality, it would be of interest if an association could be presented between the obesity and mortality among specific ischemic stroke subtypes.

Response: Thank you for the comment. Specific ischemic stroke subtypes based on TOAST criteria are not part of the structured data of GNSIS database at the present. An association between obesity and mortality among different stroke subtypes is certainly interesting and we will strive to see if it can be done in the future.

Reviewer 2 comments:

The study by Chaudhary et al. provides evidence in support of the “obesity paradox” in ischemic stroke patients. In this retrospective cohort of 6,703 first-time ischemic stroke patients, overweight and obese patients were found to have statistically decreased hazard ratio for one-year mortality in comparison to non-overweight patients. The manuscript is very well written, the methods and statistical analyses are adequately presented. The authors could consider few suggestions that might further improve the results/discussion.

1. Are underlying stroke etiologies captured in the GNSIS database? Considering the noted differences in cardiovascular risk factors in obese/overweight vs. non-overweight patients (as shown in Table 1), it would be interesting to assess whether small vessel disease or other underlying stroke etiologies might differ between obese/overweight vs. non-overweight patients (i.e., with subsequent implications in the functional recovery after stroke).

Response: We appreciate the comment. Stroke etiology and subclassification based on TOAST criteria is not part of the GNSIS database at the moment. Differences in stroke etiology between obese/overweight vs. non-overweight patients is very interesting and may be pursued in the future if possible

2. Could the authors discuss on whether ischemic/hypoxic preconditioning might be associated with improved functional outcome in obese/overweight patients? Can the rates of concomitant Obesity hypoventilation syndrome (OHS) be retrieved for included patients?

Response: Thank you for much for bringing up “ischemic/hypoxic preconditioning” as a possible explanation for improved outcome in obese/overweight patients. A section on this phenomenon has been added to the discussion [Manuscript Line 266-274].

The investigators explored Obesity Hypoventilation Syndrome (OHS) in the GNSIS database. Only 15 obese patients out of 6,703 patients included in this study had this particular diagnosis. Of these 15 patients, 5 died within 1 year, 7 were alive at 1 year and 3 were censored before 1 year. Due to the small number of patients with this diagnosis and all of them being obese, further analysis could not be done. This is partly because diagnosis of OHS requires BMI of more than 30 kg/m2 and alveolar hypoventilation which cannot be attributed to other conditions and thus this diagnosis seems to be uncommon. 

We also explored obstructive sleep apnea (OSA) rates in our study participants. OSA rate in non-overweight, overweight and obese patients was 1.4%, 3.1% and 11.5% respectively. OSA was not associated with one-year all-cause mortality and its inclusion in the stratified cox model did not change our results.

3. Minor – Line 256, p 12: “has” should be corrected to “have”.

Response: Thank you for the comment. It has been corrected in the manuscript.

We would like to thank you once again for the opportunity to strengthen our manuscript. We hope these revisions are adequate according to your requirements.

Sincerely,

Ramin Zand, MD, MPH

On behalf of co-authors

---

## [Editor Report · Decision Letter 1]

28 Jan 2021

Obesity and mortality after the first ischemic stroke: Is obesity paradox real?

PONE-D-20-39440R1

Dear Dr. Zand,

We’re pleased to inform you that your manuscript has been judged scientifically suitable for publication and will be formally accepted for publication once it meets all outstanding technical requirements.

Kind regards,

Aristeidis H. Katsanos, MD, PhD

Academic Editor

PLOS ONE
---

## [Editor Report · Acceptance letter]

1 Feb 2021

PONE-D-20-39440R1 

Obesity and mortality after the first ischemic stroke: Is obesity paradox real? 

Dear Dr. Zand:

I'm pleased to inform you that your manuscript has been deemed suitable for publication in PLOS ONE. Congratulations! Your manuscript is now with our production department. 

Kind regards, 

on behalf of

Dr. Aristeidis H. Katsanos 

Academic Editor

PLOS ONE